# Outcomes of Diabetic Retinopathy Post-Bariatric Surgery in Patients with Type 2 Diabetes Mellitus

**DOI:** 10.3390/jcm10163736

**Published:** 2021-08-22

**Authors:** Ana Maria Dascalu, Anca Pantea Stoian, Alina Popa Cherecheanu, Dragos Serban, Daniel Ovidiu Costea, Mihail Silviu Tudosie, Daniela Stana, Denisa Tanasescu, Alexandru Dan Sabau, Gabriel Andrei Gangura, Andreea Cristina Costea, Vanessa Andrada Nicolae, Catalin Gabriel Smarandache

**Affiliations:** 1Faculty of Medicine, “Carol Davila” University of Medicine and Pharmacy Bucharest, 020021 Bucharest, Romania; ana.dascalu@umfcd.ro (A.M.D.); alina.cherecheanu@umfcd.ro (A.P.C.); mihail.tudosie@umfcd.ro (M.S.T.); gabriel.gangura@umfcd.ro (G.A.G.); vanessa.nicolae@umfcd.ro (V.A.N.); catalin.smarandache@umfcd.ro (C.G.S.); 2Department of Ophthalmology, Emergency University Hospital Bucharest, 050098 Bucharest, Romania; danastanad@gmail.com; 3Department of Diabetes, Nutrition and Metabolic Diseases, “Carol Davila” University of Medicine and Pharmacy, 020021 Bucharest, Romania; ancastoian@yahoo.com; 4Fourth Department of General Surgery, Emergency University Hospital Bucharest, 050098 Bucharest, Romania; 5Faculty of Medicine, Ovidius University, 900470 Constanta, Romania; daniel.costea@365.univ-ovidius.ro; 6First Surgery Department, Emergency County Hospital, 900591 Constanta, Romania; 7ICU II Toxicology, Clinical Emergency Hospital, 014461 Bucharest, Romania; 8Fourth Department of Dental Medicine and Nursing, Faculty of Medicine, “Lucian Blaga” University, 550169 Sibiu, Romania; denisa.tanasescu@ulbs.ro; 93rd Clinical Department, Faculty of Medicine, “Lucian Blaga” University Sibiu, 550024 Sibiu, Romania; alexandru.sabau@ulbs.ro; 10Second Department of General Surgery, Emergency University Hospital Bucharest, 050098 Bucharest, Romania; 11Diaverum Nephrology and Dialysis Clinic, 900612 Constanta, Romania; acostea100@gmail.com

**Keywords:** bariatric surgery, diabetic retinopathy, progression, type 2 diabetes mellitus

## Abstract

Bariatric surgery is an emerging therapeutic approach for obese type 2 diabetes mellitus (T2DM) patients, with proven benefits for achieving target glucose control and even remission of diabetes. However, the effect of bariatric surgery upon diabetic retinopathy is still a subject of debate as some studies show a positive effect while others raise concerns about potential early worsening effects. We performed a systematic review, on PubMed, Science Direct, and Web of Science databases regarding the onset and progression of diabetic retinopathy in obese T2DM patients who underwent weight-loss surgical procedures. A total of 6375 T2DM patients were analyzed. Most cases remained stable after bariatric surgery (89.6%). New onset of diabetic retinopathy (DR) was documented in 290 out of 5972 patients (4.8%). In cases with DR at baseline, progression was documented in 50 out of 403 (12.4%) and regression in 90 (22.3%). Preoperative careful preparation of hemoglobin A1c (HbA1c), blood pressure, and lipidemia should be provided to minimize the expectation of DR worsening. Ophthalmologic follow-up should be continued regularly in the postoperative period even in the case of diabetic remission. Further randomized trials are needed to better understand the organ-specific risk factors for progression and provide personalized counseling for T2DM patients planned for bariatric surgery.

## 1. Introduction

Type 2 diabetes mellitus (T2DM) is a complex metabolic disorder characterized by insulin resistance and progressive failure of beta-pancreatic cells. The global prevalence of T2DM is estimated to rise from 8.3% in 2011 to 9.9% in 2030 [1]. The increase in T2DM incidence is well correlated with the global burden represented by a high prevalence of obesity, especially in the younger ages. Recent studies documented the link between high body mass index (BMI) and diabetes via proinflammatory cytokines, insulin resistance, increased levels of circulating fatty acids, and impaired cellular metabolism [2]. The patients newly diagnosed with T2DM are advised to lose weight and perform physical exercise to improve glycemic control and to achieve a target HbA1c below 7%, as recommended by the American Diabetes Association [3,4,5]. However, patients that lose weight by lifestyle changes, diet, or medication are not likely to maintain the results over time.

Bariatric surgery offers more rapid, efficient, and long-lasting results compared to traditional weight loss methods [5,6,7]. Current indications for bariatric surgery are a BMI of more than 40 kg/m^2^ or 35–40 kg/m^2^ associated with at least 2 obesity-related comorbidities [6,8]. Several clinical studies demonstrated that bariatric surgery in obese T2DM patients led to remission of DM in up to 75% of cases [5] by increasing insulin secretion and decreasing tissular insulin resistance. The relative impact on glucose metabolism depends on the type of surgery, being more rapid and intense in bypass intestinal procedures than in gastric restrictive interventions. Treatment during the early course of the disease ensures better outcomes before advanced beta-pancreatic cell decompensation and clinical signs of micro and macrovascular diabetic complications appear [5,9,10].

Diabetic retinopathy (DR) is one of the common microvascular complications of diabetes mellitus. According to the United Kingdom Prospective Diabetes Study (UKPDS) study, 37% of patients are diagnosed with DR and up to 60% may present some degree of it after 2 decades [11,12]. The main risk factors for the onset of DR are hyperglycemia, arterial hypertension, and hyperlipemia, leading to specific chronic microvascular changes in retinal microvasculature, which in turn lead to retinal–blood barrier disruption, leakage, ischemia, and neovascularization [13,14]. Obtaining good glycemic control is important for preventing the onset and progression of diabetic retinopathy. The UKPDS study found that patients with an HbA1c below 7% had a 21% reduction in DR progression compared to those with conventional glycemic control, while Action to Control Cardiovascular Risk in Diabetes (ACCORD) suggested that an HbA1c below 6.0 may be efficient for ensuring a significantly lower rate of progression [15,16,17]. However, there is evidence that tight glycemic control with hypoglycemic episodes may lead to an early worsening of diabetic retinopathy [18,19,20,21]. This paradoxical phenomenon is not fully understood but seems to be related to impaired autoregulation in retinal circulation that cannot adapt properly to a sudden decrease in the availability of nutrients and insulin-related increase levels of growth factors [19,20,21,22]. 

Several studies evaluated the outcomes of bariatric surgery upon diabetic retinopathy, with conflicting results. In this paper, we review the clinical evidence regarding the onset and progression of diabetic retinopathy in obese T2DM patients who underwent weight-loss surgical procedures.

## 2. Materials and Methods

Extensive research was performed on PubMed, Science Direct, Springer, and Web of Science databases, by the terms “diabetic retinopathy” AND “bariatric surgery” OR “metabolic surgery” OR “Roux-en-Y gastric bypass,” OR “sleeve gastrectomy” OR “gastric bypass” OR “biliopancreatic diversion” OR “gastric band” OR “vertical sleeve gastrectomy and their combination. For the potentially relevant records, full-text articles were obtained. Meeting abstracts, commentaries, and book chapters were excluded. Furthermore, an additional hand search was performed in the reference list of the reviews focusing on the subject. 

Original articles in English for which full texts could be obtained were included in the qualitative analyses. The strategy of research followed PICOS acronyms as recommended by PRISMA guidelines.

P: patients with T2DM

I: bariatric surgery (all types of procedures) 

C: comparison to a matched cohort of medically treated T2DM patients was analyzed when available

O: new incidence of any DR in patients with no retinopathy at baseline; % of worsening and % of improvement of DR in patients treated by bariatric surgery during the follow-up period.

S: any types of clinical studies were included.

The type of bariatric procedure used, the number of patients, follow-up time, improvement in HbA1c, and resolution of T2DM were also documented. 

### Data Extraction and Analysis

A PRISMA flowchart was employed to screen papers for eligibility. A data extraction sheet was independently completed by two researchers. We evaluated the type of study, the number of patients enrolled, the percentage of patients with DR at baseline, type of bariatric surgery, the percentage of patients who experienced de novo onset of diabetic retinopathy, progression or regression of diabetic retinopathy, and changes in HbA1c, systolic blood pressure and lipid profile. Studies providing insufficient data regarding the pre and post DR status were excluded. Any disagreement was resolved by discussion.

The risk of bias (RoB) was assessed by Egger’s test and Begg’s test. Heterogenicity of studies was analyzed by Cochran’s Q test and i^2^ test for inconsistency after the pooled effect by the random effect model (SciStat^®^ software, MedCalc Software Ltd, Ostend, Belgium).

The AMSTAR 2 (A MeaSurement Tool to Assess systematic Reviews 2) [23] chart for qualitative systematic reviews including randomized and non-randomized studies was employed to appraise the quality of the studies in the qualitative analysis. The checklist table is presented in Appendix A.

## 3. Results

The initial search resulted in 231 papers. After duplication removal and application of inclusion and exclusion criteria, 16 articles [24,25,26,27,28,29,30,31,32,33,34,35,36,37,38,39] were included in the qualitative analysis. The flowchart of the research strategy is presented in Figure 1.

The studies included in the qualitative analysis were published between 2012 and 2021, and the mean follow up ranged between 6 months and 5 years, with a median of 28.1 months. The sample size varied between 20 and 5321 participants for the surgical group, with a total of 6375 T2DM patients who underwent bariatric procedures. In 4 studies [28,35,37,38], a controlled matched group with T2DM patients treated only medically was used to compare the outcomes.

The most frequent bariatric procedures performed were: Roux-en-Y gastric bypass (RYGB), followed by laparoscopic sleeve gastrectomy (SG), laparoscopic adjustable gastric banding (GB), biliopancreatic diversion (BPD), and duodenal switch (DS).

### 3.1. Risk of Bias 

The studies included in a review were comparable in terms of selection of patients, comparison, the strategy of research, and outcomes. However, some differences may be a potential source of bias when the total change in DR status is analyzed (Egger’s test: 5.0064; *p* = 0.0014; Begg’s test: −0.1757; *p* = 0.3424). The presence of any DR and the percentages of different stages of DR at baseline varied among studies, from no DR at baseline [38] to only proliferative DR [37]. This was the reason for assessing RoB and heterogeneity of studies within subgroups, based on the analyzed effect: de novo DR, progression, or regression in DR status. The method of collecting DR data varied across the reviewed papers. Some used the retrospective data from the national health registries [38], while others documented progression by prospective [25,26,28,31,34,36] or retrospective ophthalmological evaluation [27,29,30,32,33,35,37]. The definition of progression also varied: some authors used the criteria of at least 2 steps by the ETDRS scale; others defined progression as any change in the retinopathy score.

The comparative outcomes in the reviewed studies are presented in Table 1.

In the present review, we identified 16 studies [24,25,26,27,28,29,30,31,32,33,34,35,36,37,38,39], for a total of 6375 T2DM patients that underwent bariatric surgery in which pre- and post-DR status were analyzed. The percentage of patients with no DR at baseline varied between 30.3 and 100% in the study groups. Of the total, 5972 (93.6%) patients presented no DR at baseline and 403 patients were diagnosed with various stages of DR. There was no change in the baseline DR stage in 5714 out of 6375 patients (89.6%). After bariatric surgery, 65.2% (263) of the patients with previous DR and 91.2% of the patients with no DR at presentation remained stable.

### 3.2. Incidence of De Novo DR in the Bariatric Surgery Group

New cases of DR were documented after bariatric surgery in a total of 290 of 5972 patients (4.8%) that did not present DR at baseline during the follow-up period, with a wide variation of incidence between 0 and 29.4%. The limited number of cases included in most of the available studies, limited documentation of additional risk factors (e.g., smoking) but also different follow-up periods may explain the high heterogeneity (I^2^: 91.7%) of reported outcomes (Table 2, Figure 2). For this reason, a meta-analysis could not be performed, which carried the risk of result overestimation according to the small-study effects. Risk of bias (RoB) analysis did not detect significant publication bias.

The onset of ocular microvascular complication was documented both in patients with a postsurgical total resolution of T2DM [29] and those who continued oral antidiabetic medication [24,27,28,30,32,33,34,38]. However, when the comparative incidences were compared to those of a matched group of medically treated T2DM patients, the incidence of new-onset DR was found to be significantly lower in the surgical group [38]. 

### 3.3. Progression of DR after Bariatric Surgery and Clinical Correlations

The progression of DR after bariatric surgery was documented in 50 of 403 cases (12.4%). The heterogeneity among studies was substantial (I^2^ = 75.12%), but it was lower than the de novo incidence of DR, with Egger’s (*p* = 0.3979) and Begg’s (*p* = 0.4263) tests suggesting no publication bias (Table 3, Figure 3).

Progression was detected in any stage of DR. Some authors associated advanced stages with a higher risk of progression [30], but this finding was not supported by others [29]. Kim et al. [29] found that the duration of diabetes, preexistent DR and albuminuria were risk factors for DR progression following bariatric surgery. Murphy et al. [30] found a direct correlation between initial reduction in HbA1c from pre-operative to first post-operative values and worsening of diabetic retinopathy. 

### 3.4. Regression of DR

The reviewed clinical studies documented a percentage of cases who experienced regression of DR following bariatric surgery, varying from 0 to 64% of patients with any stage of DR at baseline, with substantial heterogeneity (I^2^: 76.88%) (Table 4, Figure 4). 

In the reviewed articles, a total of 90 of 403 cases (22.3%) were documented There was no information to indicate if this positive effect could be attributed to surgery or an additional ophthalmologic therapy (laser, anti-VEGF, surgery). The improvement could be correlated with T2DM remission in the postoperative period, but also with the positive effect of bariatric surgery on other recognized risk factors for DR progression, such as hypertension and dyslipidemia.

### 3.5. The Impact of Bariatric Surgery upon Systolic Blood Pressure (SBP), Cholesterol and Serum Triglycerides (TG)

Hypertension and altered lipidic metabolism are recognized to be a risk factors for DR onset and progression, so to better document the impact of bariatric surgery on the evolution of DR in patients with T2DM, we analyzed the available data in the studies regarding the impact of bariatric surgery on blood pressure (BP), cholesterol, and triglycerides (Table 5).

When analyzing the impact of bariatric surgery upon systolic blood pressure BP, most authors found a significant decrease of systolic BP in the follow-up period [28,36,39] and a reduced necessity of antihypertensive medications [26,35]. 

All authors found a significant decrease in TG and an increase of HDL-C levels, following bariatric procedures, but with no statistically significant change in LDL-C levels [35,36,39].

Abbatini et al. [26] found a significant decrease of patients with hypertension and dyslipidemia in the study group, after a 60 months follow-up period. In the study of Thomas et al. [27], there was no information on postoperative blood pressure. However, there was a significantly higher preoperative BP mean value in the subgroup that experienced progression of DR, compared to “no change” and regression subgroups (180.3 mmHg, vs. 141.4 and 130.5, respectively). Kim et al. [29] also found an increased percentage of patients with hypertension in the group that presented progression of DR after surgery compared to the non-progression group (55 vs. 18.2%). However, this finding was not statistically significant when included in a multivariate analysis. 

## 4. Discussion

Bariatric surgery may influence the outcome of DR by multiple pathways. Achieving a target glycemic control, hypotriglyceridemia-increased insulin secretion, decreased insulin resistance, decreased inflammation and oxidative stress related to fatty tissue [40,41,42,43,44,45], and decreased blood pressure, are all well-documented factors that prevent the worsening of DR. Murphy et al. [30] and Brynskov et al. [31] raised awareness about the importance of the preoperative glycemic control to prevent a sudden decrease of HbA1c, which could be a factor for the early worsening of DR. As a consequence, DR screening for at least 5 years after bariatric surgery should be continued, particularly among those who achieved large reductions in HbA1c, as this may signal a higher risk of worsening in subsequent diabetic retinopathy screens [30]. However, in long-term follow-up, Chang et al. [46] found no differences between pre and post-surgery HbA1c in the progression vs. non-progression group. Moreover, though there is evidence that RYGB induces an earlier and more important decrease in HbA1c, Amin et al. [32], could not find a significantly higher risk for DR progression in a comparative study of bariatric procedures.

Several large retrospective studies [46,47,48,49] investigated the long-term risk of microvascular complications in T2DM patients after bariatric surgery compared to medical treatment, and found a significant decrease in microvascular events: a reduced cumulative incidence of DR by 45% [47], a decreased incidence of STDR by 42% [46] and a lower risk of blindness and need for laser or ocular surgery [48] compared to medical treatment. Moreover, Carlsson et al. [49] found that those who achieved remission after surgery had a significantly lower incidence of microvascular events compared with those who were not in remission after 15 years (8.0 versus 25.2 events per 1000 person-years). Amin et al. [32] found a lower progression to STDR or maculopathy in patients with T2DM following bariatric surgery compared to routine care. All this clinical evidence reflects the beneficial metabolic effects of bariatric surgery [32].

On the other hand, Schauer [50] and Chang [51] found no significant differences in changes to the DR score in the follow-up period. One explanation may be the limited number of patients included in the study group.

Only one study analyzed the outcomes of bariatric surgery in a group of patients with proliferative DR baseline [37] and found an increased risk of complications, such as intraocular hypertension, neovascular glaucoma, and retinal vein thrombosis in surgical vs. medically treated matched patients. However, the authors acknowledged that the patients that underwent bariatric surgery missed their ophthalmological check-ups within the first 3 months, thus having a lower number of intravitreal anti-VEGF injections compared to the medically treated group. Whether this finding could be a contributor to increased complications should be analyzed in further studies.

In the present review, we found that, in most cases, DR remained stable after bariatric procedures. We also found a progression of retinal lesions in 12.4% of cases and regression in 22.3%. In 4.8% of cases with no retinopathy at baseline, retinal microvascular changes appeared after bariatric surgery, despite achieving a target glycemic control. 

There are some limitations to the present review: the small number and the substantial heterogeneity of the studies, the lack of information about different methods to assess DR, the different treatments proposed in cases of severe DR either before or after bariatric surgery, and other important factors known to influence RD status, such as smoking, drug use, physical activity and nutrition. Moreover, the probable direct influence on DR of pre-surgical global management (physical activity, nutrition, motivational interviewing regular and frequent follow up by several health professionals) generally went from 6 to 12 months before the surgery procedure. In a long-term follow-up, other factors could influence the outcome more.

However, clinicians should be aware that there is a change of progression, despite even T2DM remission after surgery. The mechanisms involved are still a subject of research. Complex hormonal and metabolic changes following bariatric surgery may be responsible for the rapid improvement of glycemic control even before weight loss is achieved, and may be a cause for the early worsening of DR. Other factors for DR progression may involve vitamin and microelement deficiencies (A, D, B12, copper, and folate) which are secondary to malnutrition or the discontinuation of oral antidiabetic medication associated with a protective role for retinal damage, such as fenofibrate or angiotensin receptor blockers (ARBs) [36,52,53,54,55].

### 4.1. Gut Hormones and Metabolic Changes after Bariatric Surgery Procedures

Bariatric surgery nowadays offers a different therapeutic approach to T2DM associated with obesity. The intrinsic mechanisms responsible for diabetes resolution are still a subject of research. Several studies revealed a 25–47% early remission of diabetes after gastric restrictive procedures, up to 45–90% after Roux-en-Y anastomosis [37,49,56,57] and up to 95% following biliopancreatic derivation [49].

Weight loss is associated with negative energy balance and decreased tissular insulin resistance. On the other hand, the decrease in the amount of fat tissue reduces oxidative stress and chronic inflammation associated with lipotoxicity, including fatty acids and chemokines secreted by adipocytes [45,58].

The bariatric procedures may be classified according to the mechanism of weight loss in gastric restrictive and intestinal by-pass procedures. The gastric restrictive procedure uses approaches, such as a gastric sleeve or laparoscopic adjustable gastric banding [5] to reduce gastric volume, induce early satiety, and reduce caloric intake [59]. Moreover, removal of the gastric fundus decreases the level of ghrelin, thereby causing decreased hunger [59,60].

Intestinal bypass procedures, such as Roux-en-Y gastric bypass and biliopancreatic diversion have a deep impact on gastrointestinal physiology. By shortening the small intestine and altering the alimentary pathway in the digestive tract, they reduce the gastric reservoir and cause the selective malabsorption of fats and other nutrients [5,61,62]. The associated dumping syndrome is expected to limit patients’ consumption of triggering food, such as those high in sugars. 

The gastrointestinal tract has a complex role in metabolic regulation. The hindgut theory formulated by Cumming et al. [46] is based on the fact that the rapid delivery of incompletely digested food to the lower intestinal tract is triggers the secretion of incretins, such as gastric inhibitory polypeptide (GIP) and glucagon-like peptide 1 (GLP-1), thus increasing the secretion of glucose-dependent insulin, suppressing glucagon, and inducing satiety. Peptide YY, normally secreted in the distal small bowel and responsible for satiety in response to food intake, is expressed earlier and at higher levels, especially in intestinal bypass procedures [63]. 

Rubino et al. formulated in 2002 the foregut hypothesis, showing that bypassing a short segment of the proximal intestine directly ameliorates type 2 diabetes, independently of the effects on food intake, body weight, malabsorption, or nutrient delivery to the hindgut [64]. In this way, a dysfunctional signal released by nutrient contact with the proximal duodenum, including the release of “anti-incretins” that lead to insulin resistance and T2DM, is inhibited [64,65].

Intestinal bypass procedures act aggressively on lowering blood sugar, even from the first week before weight loss occurs, via postprandial GLP-1-mediated insulin secretion. Recent studies found that GLP-1 acts on multiple levels, by decreasing hepatic glucose synthesis and regulating food intake by synergistic actions at the brain and gut levels [66,67]. Gastric restrictive procedures are considered to have a milder effect on glucose metabolism [68,69,70]. However, no study included in the review found significant differences between the type of surgery and risk for DR progression in a 12-month or longer follow-up.

### 4.2. The Paradoxical Effect of Glucose-Lowering Therapy on Diabetic Retinopathy 

The retina is one of the most metabolically active tissues in the body because neural–retinal energy is exclusively represented by glucose [71]. The severity of DR at baseline is associated with decreased autoregulation in retinal microcirculation and increased vulnerability to osmotic or nutrient concentration changes [72]. Early worsening of DR was first documented in a Diabetes Control and Complications Trial (DCCT) with intensive glycemic lowering therapy [20]. Afterwards, clinical evidence was observed with the beginning of insulin therapy and GLP-1 analogs [73,74,75]. Similar concerns were also raised regarding bariatric surgery. 

The biochemical mechanism by which hypoglycemia and high insulin levels can induce retinal damage is not completely understood [76,77]. A decreased level of blood sugar triggers osmotic changes that favor leakage and retinal edema from previously damaged capillaries. Low levels of oxygen and glucose are potential triggers for VEGF retinal production [22].

Experimental studies in vitro showed a synergistic interaction between insulin and VEGF on blood vessels on endothelial bovine cells, a disruption of the blood–retinal barrier, increased ROS [78], and expression of hypoxia-inducible factor (HIF-1) ion diabetic rats following intensive insulin therapy [22,79] with subsequent inflammation and neovascularization.

### 4.3. DR Phenotypes and Risk for Progression

Classically, HbA1c was considered a powerful clinical tool for predicting the progression of diabetic retinopathy [80,81]. However, clinical observations showed that there are patients who maintain good vision despite poor glycemic control, and vice-versa. Recent research characterized 3 DR phenotypes associated with different dominant retinal alterations and different risks of progression to vision-threatening complications [82]. Microaneurysm turnover was defined as the number of new microaneurysms and vanished microaneurysms over a preset period of 6 months as evaluated by fundus photography with or without retinal angiography. Microaneurysm (MA) turnover and macular thickness measured by optical coherence tomography may be considered organ-specific markers for predicting the long-term outcomes and maybe suggesting the predominance of a different pathogenic mechanism. Cunha-Vaz et al. [83,84] defined phenotype A, with a MA turnover of <6 and normal retinal thickness, characterized by apoptosis, vessel regression, and loss of pericytes, with limited progression over time. Phenotype B was characterized by MA turnover <6 but associated with an increased retinal thickness measured by OCT, which should suggest a predominant role of blood–retinal barrier disruption and inflammation. Phenotype C, associated with a MA turnover > 6 and variable retinal thickness, was characterized as having a higher risk for visual impairment due to ischemia and angiogenesis. Further clinical validation of these different phenotypes may lead to the development of targeted treatments and personalized approaches in the management of DR [83,84,85]. 

## 5. Conclusions

Although diabetic retinopathy in most cases is stable after bariatric surgery, there is still little information about the prediction of DR outcomes in individuals in remission, cured or with T2DM. The careful preoperative preparation of HbA1c, blood pressure, and lipidemia should be provided to minimize the expectation of DR worsening. Ophthalmologic post-operative follow-ups should be continued regularly, despite diabetic remission for early detection and treatment of possible diabetic retinopathy progression. Further randomized trials are needed to better understand the organ-specific risk factors for progression and provide adequate personalized counseling to the T2DM patients that are planned to have bariatric surgery.

## Figures and Tables

**Figure 1 jcm-10-03736-f001:**
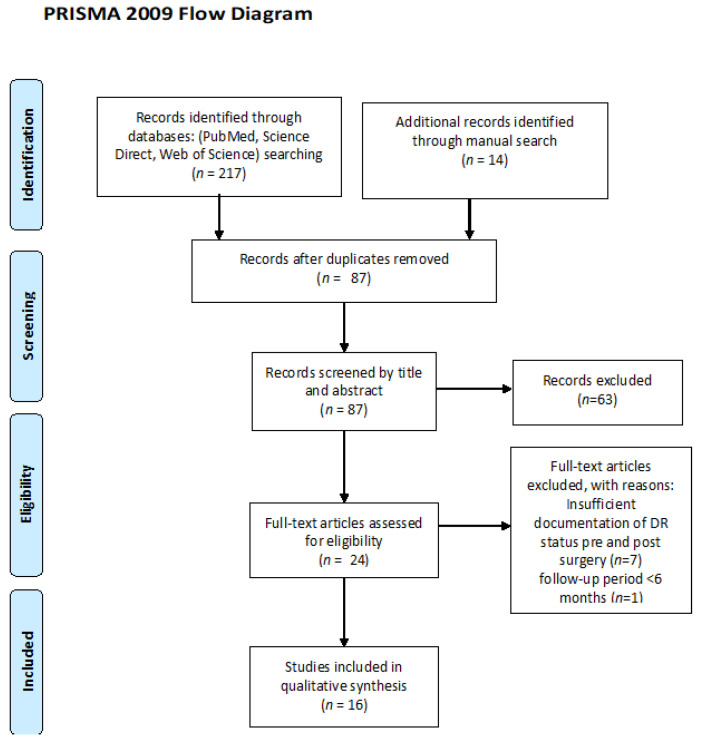
PRISMA flowchart for the studies included in the review. DR: diabetic retinopathy.

**Figure 2 jcm-10-03736-f002:**
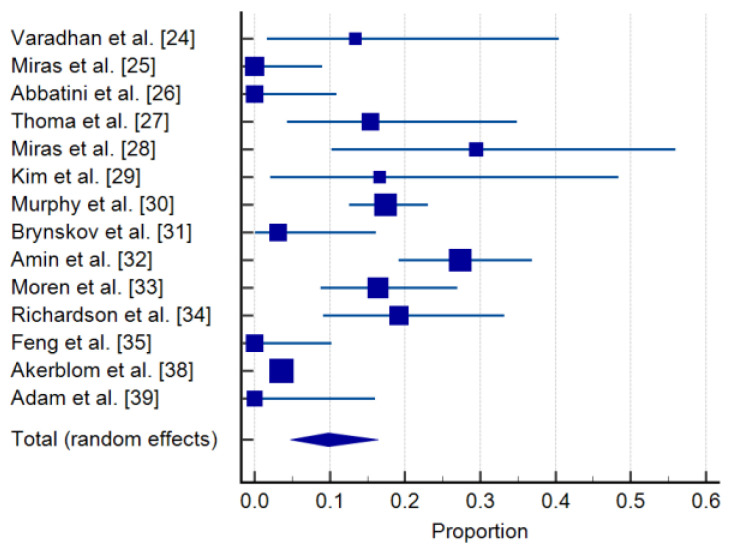
Incidence of de novo DR in screened studies. Forrest plot: pooled effects-random effects model [24,25,26,27,28,29,30,31,32,33,34,35,38,39].

**Figure 3 jcm-10-03736-f003:**
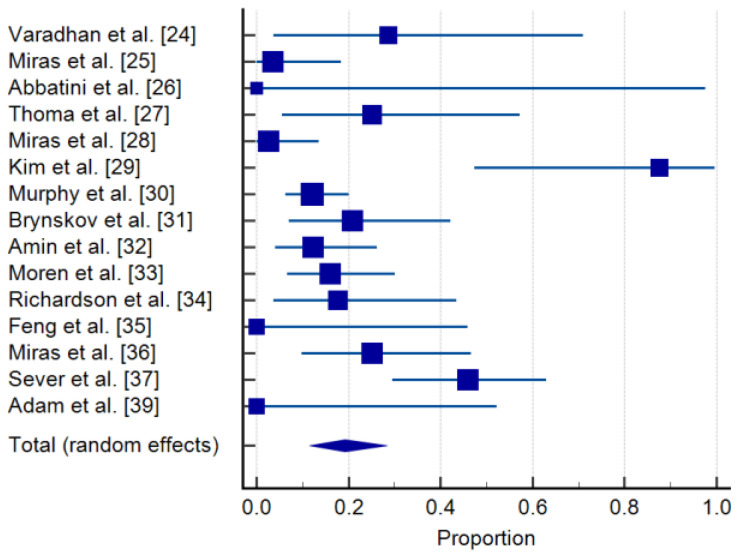
Progression of DR in patients with DR at baseline. Forrest plot: pooled effects–random effects model [24,25,26,27,28,29,30,31,32,33,34,35,36,37,39].

**Figure 4 jcm-10-03736-f004:**
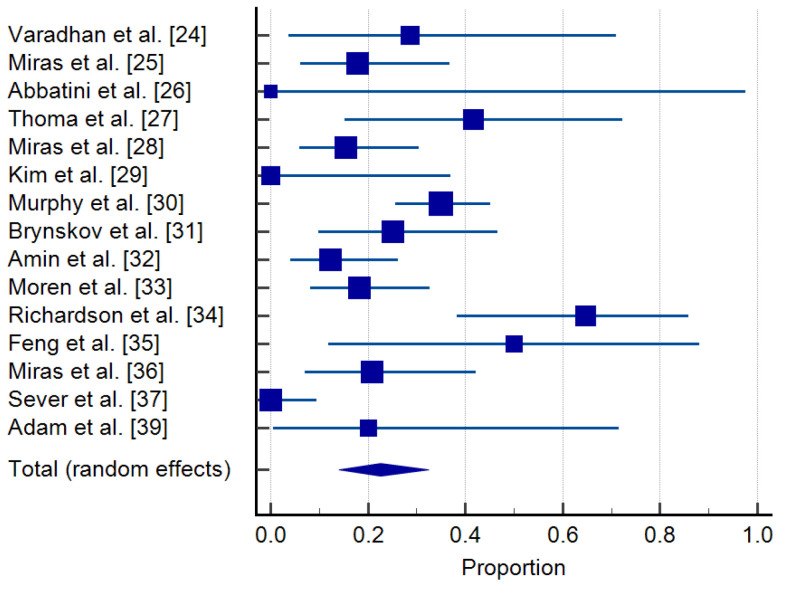
Regression of DR in patients with DR at baseline. Forrest plot: pooled effects–random effects model [24,25,26,27,28,29,30,31,32,33,34,35,36,37,39].

**Table 1 jcm-10-03736-t001:** Details of the studies included in the systematic review.

Study, Year	No of Patients (Surgical; Medical)	Type of Study	Follow-Up Period (Months)	Baseline No DR/DR	Type of Bariatric Procedure	New Onset of DR	% DR Worsening	%DR Improving	% No Change in DR Stage	Change in HbA1c (%)	Discontinuation of Oral Medication
Varadhan et al., 2012 [24]	22	retrospective	6–12	15/7	SG, RYGB	2/15 (13%)	2/7 (9%)	2/7 (9%)	16/22 (73%)	2.0% (0.3–4.2%)	No info
Miras et al., 2012 [25]	67	prospective	12–18	39/28	SG 22.6% RYGB 70.2% GB 7.1%	0/39	1/28(3.6%)	5/28 (17.8%)	61/67 (91%)	No info	No info
Abbatini et al., 2013 [26]	33	prospective	36–60	32/1	SG	0/32	0/1	0/1	33/33	−2.0%	76.9%
Thomas et al., 2014 [27]	38	retrospective	12	26/12	SG 35% RYBP 30% GB 10% BPD 25%	4/26 (15%)	3/12 (25%)	5/12 (42%)	26/38 (68%)	−1.5%	No info
Miras et al., 2015 [28]	56; 21	prospective	12	17/39 9/12	RYGB vs. med	5/17; 0/9	1/39 (2.5%) 3/12 (25%)	6/39 (15%) 1/12 (8.3%)	44/56(78%) 17/21(81%)	−3.3% +0.7%	decreased medication by 41% medication vs. increased medication by 27%
Kim et al., 2015 [29]	20	retrospective	12–46	12/8	RYGB	2/12 (16.6%)	7/8 (87.5%)	0%	11/20 (55%)	−2.4%	6(30%) remission T2DM; 3 cases experienced DR progression
Murphy et al., 2015 [30]	318	retrospective	12	218/100	RYGB 30.8% SG 65.7% DS 3.5%	38/218 (17%)	12/100 (12%)	35/100 (35%)	232/318 (73%)	−3.9%	18%
Brynskov et al., 2016 [31]	56	prospective	12	32/24	RYGB 94% SG 6%	1/32 (3%)	Any visit: 5/24 (21%) 12 mo: 3/24 (13%)	Any visit: 6/24 (25%) 12 mo: 4/24 (17%)	49/56 (87%)	−1.7%	59%
Amin et al., 2016 [32]	152	Retrospective cohort analysis	36	106/41	GB 70% RYGB 25% SG 4.6%	29/106 (27%)	5/41 (12%)	5/41 (12%)	113/152 (74%)	−0.9%	*n*/a
Moren et al., 2018 [33]	117	retrospective	16	73/44	RYGB	12/73 (12%)	7/44 (16%)	8/44 (18%)	90/117 (77%)	−1.9%	66%
Richardson et al., 2018 [34]	32 (64 eyes)	prospective	36	47/17	RYGB	9/47 (19%)	3/17 (17%)	11/17 (64%)	41/64 (64%)	*n*/a	*n*/a
Feng et al., 2019 [35]	40; 36	Retrospective controlled	12	34/6 29/7	RYGB Vs med	-	-	3/34 (8%) vs. 0%	37/40 (92%) vs. 100% 36/36 (1005)	−1.9% −0.3%	48 vs. 3%
Miras et al., 2019 [36]	24	prospective	60	*n*/a	RYGB, SG, GB	0/24	6/24 (25%)	5/24 (20.8%)	13/24 (54.2%)	−1.4%	43%
Sever et al., 2020 [37]	21 (37 eyes) 27 (37 eyes)	Retrospective, comparative	12	PDR only	*n*/a	Increased % of complication in surgical vs. medical group: IOH, NVG, retinal vein occlusion (21.6, 16, 8 vs. 5.4, 2.7, 0)				−1.0% −0.7%	*n*/a
Akerblom et al., 2021 [38]	5321; 5321	Retrospective database analysis, comparative	54	No DR only	RYGB	188 (3.5%) 317 (5.9%)	-	-	5133 (94.3%) 5004 (94.1%)	*n*/a	*n*/a
Adam et al., 2021 [39]	26	prospective	12	21/5 (R1)	RYGB-21 (81) SG-5 (19%)	0%	0%	1(20%)	25/26 (96%)	−1.4%	*n*/a

DR: Diabetic retinopathy; HbA1c: Hemoglobin A1c; SG: sleeve gastrectomy; RYGB: Roux-en-Y gastric bypass; GB: Gastric banding; BPD: biliopancreatic derivation; DS: duodenal switch; STDR: sight-threatening DR; NPDR: non-proliferative DR; PDR: proliferative DR; IOH: intraocular hypertension; NVG: neovascular glaucoma.

**Table 2 jcm-10-03736-t002:** Incidence of de novo DR in screened studies: means and 95% credible intervals.

Study	Sample Size	Proportion (%)	95% CI	Weight (%)
				Random
Varadhan et al. [24]	15	13.333	1.658 to 40.460	5.75
Miras et al. [25]	39	0.000	0.000 to 9.025	7.32
Abbatini et al. [26]	32	0.000	0.000 to 10.888	7.05
Thomas et al. [27]	26	15.385	4.356 to 34.868	6.73
Miras et al. [28]	17	29.412	10.314 to 55.958	5.98
Kim et al. [29]	12	16.667	2.086 to 48.414	5.31
Murphy et al. [30]	218	17.431	12.640 to 23.131	8.60
Brynskov et al. [31]	32	3.125	0.0791 to 16.217	7.05
Amin et al. [32]	106	27.358	19.149 to 36.874	8.26
Moren et al. [33]	73	16.438	8.793 to 26.954	7.99
Richardson et al. [34]	47	19.149	9.149 to 33.260	7.55
Feng et al. [35]	34	0.000	0.000 to 10.282	7.13
Akerblom et al. [38]	5321	3.533	3.053 to 4.065	8.93
Adam et al. [39]	21	0.000	0.000 to 16.110	6.37
Total (random effects)	5993	9.818	4.784 to 16.397	100.00

Test for heterogeneity: Q: 156.56; DF: 13; *p* < 0.0001; I^2^: 91.7%; Publication bias: Egger’s test: 2.1131 (*p* = 0.0521); Begg’s test: Kendall’s Tau: −0.04420 (*p* = 0.8257).

**Table 3 jcm-10-03736-t003:** Progression of DR in patients with DR at baseline: means and 95% credible intervals.

Study	Sample Size	Proportion (%)	95% CI	Weight (%)	
				Random	
Varadhan et al. [24]	7	28.571	3.669 to 70.958	4.93	
Miras et al. [25]	28	3.571	0.0904 to 18.348	7.86	
Abbatini et al. [26]	1	0.000	0.000 to 97.500	1.94	
Thomas et al. [27]	12	25.000	5.486 to 57.186	6.14	
Miras et al. [28]	39	2.564	0.0649 to 13.476	8.38	
Kim et al. [29]	8	87.500	47.349 to 99.684	5.23	
Murphy et al. [30]	100	12.000	6.357 to 20.024	9.37	
Brynskov et al. [31]	24	20.833	7.132 to 42.151	7.58	
Amin et al. [32]	41	12.195	4.081 to 26.204	8.45	
Moren et al. [33]	44	15.909	6.644 to 30.065	8.55	
Richardson et al. [34]	17	17.647	3.799 to 43.432	6.90	
Feng et al. [35]	6	0.000	0.000 to 45.926	4.59	
Miras et al. [36]	24	25.000	9.773 to 46.711	7.58	
Sever et al. [37]	37	45.946	29.487 to 63.078	9.31	8.30
Adam et al. [39]	5	0.000	0.000 to 52.182	1.47	4.20
Total (random effects)	393	19.231	11.554 to 28.315	100.00	100.00

Test for heterogeneity: Q: 56.267; DF: 14; *p* < 0.0001); I^2^: 75.12% (95% CI: 58.82 to 84.97); Publication bias: Egger’s test: 1.1090 (*p* = 0.3979); Begg’s test: Kendall’s Tau: 0.1531 (*p* = 0.4263).

**Table 4 jcm-10-03736-t004:** Regression of DR in patients with DR at baseline: means and 95% credible intervals.

Study	Sample Size	Proportion (%)	95% CI	Weight (%)
				Random
Varadhan et al. [24]	7	28.571	3.669 to 70.958	5.04
Miras et al. [25]	28	17.857	6.064 to 36.893	7.81
Abbatini et al. [26]	1	0.000	0.000 to 97.500	2.04
Thomas et al. [27]	12	41.667	15.165 to 72.333	6.21
Miras et al. [28]	39	15.385	5.862 to 30.528	8.29
Kim et al. [29]	8	0.000	0.000 to 36.942	5.33
Murphy et al. [30]	100	35.000	25.729 to 45.185	9.19
Brynskov et al. [31]	24	25.000	9.773 to 46.711	7.56
Amin et al. [32]	41	12.195	4.081 to 26.204	8.36
Moren et al. [33]	44	18.182	8.192 to 32.714	8.44
Richardson et al. [34]	17	64.706	38.328 to 85.790	6.92
Feng et al. [35]	6	50.000	11.812 to 88.188	4.71
Miras et al. [36]	24	20.833	7.132 to 42.151	7.56
Sever et al. [37]	37	0.000	0.000 to 9.489	8.22
Adam et al. [39]	5	20.000	0.505 to 71.642	4.33
Total (random effects)	393	22.568	14.040 to 32.441	100.00

Test for heterogeneity: Q: 60.5611; DF: 14; *p* < 0.0001); I^2^: 76.88%; Publication bias: Egger’s test: 0.097 (*p* = 0.9439); Begg’s test: Kendall’s Tau: 0.1723 (*p* = 0.3708).

**Table 5 jcm-10-03736-t005:** The impact of bariatric surgery upon BP, LDL-C, HDL-C, and TG.

Study	Preop SBP Mean ± DS (mmHg)	Postop SBP (Mean ± DS), *p*-Value	Preop Cholesterol (mg/dL)	Postop Cholesterol (mg/dL), *p* Value	Preop TG	Postop TG, *p* Value
Abbatini et al., 2013 [26]	-	% patients with hypertension decreased from 54.5% to 15.1%	-	% patients with hypercholesterolemia decreased from 21 to 9%	-	% patients with hyperTG decreased from 18 to 9%
Thomas et al., 2014 [27]	DR progression: 180.3 ± 32.8 DR no change:141.4 ± 17.0 DR regression: 130.5 ± 27.6	No info	181.7 (38.7)	166.3 (*p* = 0.36)	No info	No info
Miras et al., 2015 [28]	143 ± 2	130 ± 3 (*p* < 0.001)	No info	No info	No info	No info
Feng et al., 2019 [35]	134.0 ± 3.6	123.1 ± 2.9 (*p* < 0.05) complete remission in 14/24 (58%)	193.35 ± 7.73	154.68 ± 7.73, *p* < 0.001	265.7 ± 35.43	97.43 ± 8.857 *p* < 0.001
Miras et al., 2019 [36]	142 (103–195)	128 (104–196) *p* < 0.0001	Total C: 181.75 HDL-C: 42.54 LDL-C: 100.54	Total C: 170.15, *p* = 0.21 HDL-C: 54.14, *p* < 0.0001 LDL-C: 88.94, *p* = 0.16	159.43 (53.14–655.4)	115.15 (35.43–389.7), *p* < 0.0001
Adam et al., 2021 [39]	134 ± 15	119 ± 15, *p* < 0.001	Total C: 144 ± 28.6 HDL-C: 33.2 (29.7–39.0) LDL-C: 81.9 ± 23.9	Total C: 162 ± 36.7, *p* = 0.035 HDL-C: 44.0 (38.6–50.6), *p* < 0.001 LDL-C: 93.8 ± 35.1, *p* = 0.38	134 (81.4–165)	100 (77.0–132), *p* = 0.071

SBP: systolic blood pressure; total C: total cholesterol; LDL-C: low-density lipoprotein cholesterol; TG and Cholesterol values converted from mmol/L.

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
