# Peer review of "Outcomes of Diabetic Retinopathy Post-Bariatric Surgery in Patients with Type 2 Diabetes Mellitus"

_jcm, 2021, doi:10.3390/jcm10163736_

Round 1

Reviewer 1 Report

Dear authors,

Thank you for the opportunity to review your Systematic review investigating post-bariatric surgical outcomes for Diabetic retinopathy. I found the review interesting and the introduction and discussion sections were well written. I recommend a review of typos and language generally as there were a number of minor errors throughout. The methods section was very brief and would benefit from elaboration on several items. I have some additional comments and considerations for your manuscript below.

  1. The introduction was appropriate
  2. The 'Risk of bias' section on Pg 3, appears to be a summary of the included articles and would fit best into the results section. This section would also benefit from references of included articles, as appropriate,  for each statement. 
  3. The methods requires a statement of which quality appraisal tool was used (if any). Ideally a Quality appraisal table should be provided if the journal has capacity for an additional table or as supplementary material.
  4. I could not see any information as to the search strategy. Specifically, how many people ran the search or performed screening at abstract and full-text level, for instance?
  5. You could also elaborate on the search terms further to enable to search to be re-produced.
  6. There were some instances of discussion creeping into the 'results' section. Please be succinct in the results, only presenting the main findings without discussion.
  7. The discussion was quite well written and appeared justified, as was the conclusion.

Author Response

Dear Reviewer,

Thank you for appreciating our research and for your valuable comments.

We have carefully revised our manuscript, according to your recommendations and checked the English typing errors. We expanded the Materials and method section, describing in detail how the research was performed.

  1. The 'Risk of bias' section on Pg 3, appears to be a summary of the included articles and would fit best into the results section. This section would also benefit from references of included articles, as appropriate,  for each statement. 

We moved this section into the results section, adding the references, as required.

  1. The methods requires a statement of which quality appraisal tool was used (if any). Ideally a Quality appraisal table should be provided if the journal has capacity for an additional table or as supplementary material.

We have added a paragraph regarding the quality appraisal tool used, and add the AMSTAR2 checklist as a supplementary file.

  1. I could not see any information as to the search strategy. Specifically, how many people ran the search or performed screening at abstract and full-text level, for instance?

We expanded the Materials and method section, describing in detail how the research was performed.

  1. You could also elaborate on the search terms further to enable to search to be re-produced.

We have added more details about the search terms and how the research was performed. As well, we added an evaluation regarding the heterogenicity of the studies (Q test and i2 test) and potential publication bias (Egger’s test and Begg’s test), based on the other reviewer’s recommendations.

  1. There were some instances of discussion creeping into the 'results' section. Please be succinct in the results, only presenting the main findings without discussion.

We have shortened the Results section to make it more succinct and detailed the data obtained in the discussion section

We hope in this revised version you will find it suitable to be published.

Kind regards,

Assoc. Prof. Dr. Dragos Serban

Reviewer 2 Report

In Methods:

Since the aim of the study is to assess the Outcomes of Diabetic Retinopathy post Bariatric Surgery, I believe including studies without DR status at baseline,  in this review is inappropriate.

Indeed, to be able to talk about Incidence of “de novo” DR or Progression of DR after bariatric surgery, baseline status of DR appears mandatory. Thus, the studies with the lack of these informations should be removed from the analysis. And finally, only the 16 studies cited in lines 158-159 respect the inclusion and exclusion criteria.

In the results:

the heterogeneity of the studies might be calculated and mentioned.

In the discussion:

A paragraph summarizing the severals limits of this work should be mentioned before the discussion of the physiopathologic mechanisms of DR.

Among the limitations of this work:

  • the small number and the probable high heterogeneity of the studies
  • the lack of information about the different methods to assess RD and the different treatments proposed in case of severe RD either before or after bariatric surgery
  • the other important factors known to influence RD status (smoking status, drugs, physical activity, nutrition)
  • Moreover the probable direct influence on RD of the pre-surgical global management (physical activity, nutrition, motivational interviewing regular and frequent follow up by the several health professionals) generally going from 6 to 12 months before the surgery procedure
  • Authors may mention that to date, no specific or dedicated study that aimed to assess the RD status and progression after bariatric surgery as a primary outcome, exists.

Author Response

Dear reviewer,

Thank you very much for your time and your valuable comments. We have revised the manuscript according to your recommendations.

In Methods:

Since the aim of the study is to assess the Outcomes of Diabetic Retinopathy post Bariatric Surgery, I believe including studies without DR status at baseline,  in this review is inappropriate.

Indeed, to be able to talk about Incidence of “de novo” DR or Progression of DR after bariatric surgery, baseline status of DR appears mandatory. Thus, the studies with the lack of these informations should be removed from the analysis. And finally, only the 16 studies cited in lines 158-159 respect the inclusion and exclusion criteria.

We have removed the 5 studies lacking detailed data about DR status and left only the 16 studies cited in lined 158-159, as recommended.

In the results:

the heterogeneity of the studies might be calculated and mentioned.

We have assessed the heterogenicity of the studies (Q test and i2 test) and potential publication bias (Egger’s test and Begg’s test) for the 16 studies – for total change in DR status, and also for effect on subgroups – “de novo” DR; progression of DR; regression of DR.

In the discussion:

A paragraph summarizing the several limits of this work should be mentioned before the discussion of the physiopathologic mechanisms of DR.

Among the limitations of this work:

  • the small number and the probable high heterogeneity of the studies
  • the lack of information about the different methods to assess RD and the different treatments proposed in case of severe RD either before or after bariatric surgery
  • the other important factors known to influence RD status (smoking status, drugs, physical activity, nutrition)
  • Moreover the probable direct influence on RD of the pre-surgical global management (physical activity, nutrition, motivational interviewing regular and frequent follow up by the several health professionals) generally going from 6 to 12 months before the surgery procedure
  • Authors may mention that to date, no specific or dedicated study that aimed to assess the RD status and progression after bariatric surgery as a primary outcome, exists.

We added in the Discussion section a paragraph regarding the limitations of the study. Thank you very much for your valuable suggestions.

Kind regards,

Assoc. Prof. Dr. Dragos Serban